# Exploring the variation in muscle response testing accuracy through repeatability and reproducibility

Anne M. Jensen [1,2,3]*, Richard J. Stevens[1,2], Amanda J. Burls[4]

1 Department of Continuing Professional Education, University of Oxford, Oxford, United Kingdom,
2 Department of Primary Care Health Sciences, University of Oxford, Oxford, United Kingdom,
3 HeartSpeak International, Brisbane, Queensland, Australia, 4 School of Health Sciences, City University London, London, United Kingdom

* dranne@drannejensen.com

## Abstract

### Research Objectives

To explore the variation in mean muscle response testing (MRT) accuracy and whether this variation can be attributable to participant characteristics.

### Methods

A prospective study of diagnostic test accuracy was carried out in a round-robin format. Sixteen practitioners tested each of 7 test patients (TPs) using 20 MRTs broken into 2 blocks of 10 which alternated with 2 blocks of 10 intuitive guessing. Mean MRT accuracies (as overall percent correct) were calculated for each unique pair. Reproducibility and repeatability were assessed using analyses of variance (ANOVA) and scatter and Bland-Altman plots.

### Results

The mean MRT accuracy (95% CI) was 0.616 (0.578–0.654), which was significantly different from both the mean intuitive guessing accuracy, 0.507 (95% CI 0.484–0.530; p<0.01) and chance (p<0.01). Visual inspection of scatterplots of mean MRT accuracies by practitioner and by TP suggest large variances among both subsets, and regression analysis revealed that MRT accuracy could not be predicted by TP (r=–0.14; p=0.19), nor by Practitioner (r=0.01; p=0.90). A significant effect imposed by both practitioners and TPs individually and together was found at the p<0.05 level; however, together they account for only 57.0% of the variance, with 43.0% of the variance unexplained by this model. From a statistical perspective, Bland-Altman Plots of mean MRT accuracy by practitioner do show adequate repeatability since all scores fell within 2 SDs of the mean; however, the wide range of scores also suggests insufficient repeatability from a clinical perspective. Finally, ANOVA

**Data availability statement:** All relevant data are within the manuscript and its Supporting Information files.

**Funding:** The author(s) received no specific funding for this work.

**Competing interests:** The authors have declared that no competing interests exist.

demonstrated that an insignificant amount of variance could be explained by block [$F(1,21) = 0.02$, $p = 0.90$].

---

## 1. Introduction

Muscle response testing (MRT), one type of manual muscle testing (MMT), is used predominantly by complementary and alternative health care providers, including osteopaths, physiotherapists, chiropractors, kinesiologists, naturopaths and psychologists. The objective of MRT goes beyond that of MMT whose aim is to evaluate neuromusculoskeletal integrity. That is, in MRT, muscles are tested, not to evaluate muscular strength (as in MMT) [1], but to detect the presence of and contributing factors toward target conditions, such as low back pain [2], simple phobia [3,4], and food allergies [5]. MRT is differentiated from other types of MMT since typically only one muscle is used for testing, and is tested repeatedly, as the target condition changes [6].

Because MRT is estimated to be used by over 1 million people worldwide [7], assessing its validity is crucial. According to Bossuyt, the first question to ask in the evaluation a new diagnostic test is "Does it measure what it is supposed to measure?" – otherwise known as its analytic validity [8]. Moreover, the first step in assessing a test's analytic validity is to estimate its accuracy, which can be measured as sensitivity and specificity, overall fraction correct, positive predictive value and negative predictive value and others. This study is one in a series of studies investigating the validity of MRT used to distinguish lies from truth [6,9–12]. Previous studies in the series estimated the accuracy of MRT to be in the 60–70% correct range which was significantly different to change or guessing [6,9,11,12]. Since a diagnostic test is only considered valid if it is both accurate *and* precise, assessing the precision of MRT is an important next step.

Because of the ambiguity found in previous studies of the validity of muscle testing, it is important to be clear about the definition of terms used in this study. Precision can be defined as "the degree to which repeated measurements under unchanged conditions show the same results" [13]. Just as there are numerous ways to quantify the accuracy of a diagnostic test [8], there are several terms currently used to describe its precision, such as reproducibility, repeatability, reliability (inter-tester and intra-tester), and stability. However, it is most common to gauge the precision of a test in terms of its reproducibility and repeatability [14]. Unfortunately, these two terms are also frequently confused. For clarity, they are defined as:

Reproducibility: the variability of the average values obtained by several testers while measuring the same item (inter-tester variability) [14]

Repeatability: the variability of the measurements obtained by one person while measuring the same item repeatedly (intra-tester variability) [14]

Applying these terms to the context of MRT, reproducibility, then, may be described as *the degree of variability in MRT accuracy between different practitioners testing the same test patient (TP)*, and repeatability may be described as *the degree of variability in MRT accuracy when a practitioner tests the same TP at different times*.

Hence, the aim of this study was to assess the precision of MRT used for distinguishing lies from truth, so that that both reproducibility and repeatability could be evaluated. More specifically, the research questions for this study were: [1] Is the MRT accuracy that a practitioner achieves with one TP consistent over many TPs, or is it TP- or pair-specific?; and [2] Is the MRT accuracy obtained with one TP consistent over many practitioners, or is it practitioner- or pair-specific?

## 2. Methods

The protocol for this prospective study of diagnostic test accuracy received ethics committee approval by the Oxford Tropical Research Ethics Committee (OxTREC; Approval #41–10) and the Parker University Institutional Review Board for Human Subjects (Approval # R16_10). Also, this study protocol was registered with two clinical trials registries: US-based ClinicalTrials.gov, and the Australian New Zealand Clinical Trials Registry (ANZCTR; www.anzctr.org.au). All participants gave their written informed consent, and all other tenets of the Declaration of Helsinki were upheld. This paper followed the Standards for the Reporting of Diagnostic Test Accuracy Studies (STARD) guidelines (see S1 Table, for the STARD Checklist) [15–17].

### 2.1. Participants and setting

Volunteer practitioners were solicited from a group of muscle testing practitioners attending a seminar in Dallas, Texas, on 26 June 2012. All practitioner recruitment, all enrollment and all data collection were done during the course of one day. In addition, seven [7] TPs were recruited and enrolled in the few days leading up to the event from a convenience sample of a mixture of MRT-naïve and non-MRT-naïve individuals. Participants were sought who were aged 18–65 years, had fully functioning and painfree upper extremities, and were fluent in English. Volunteers were excluded if they were markedly hearing-, sight- or speech-impaired.

Once enrolled, practitioners waited in a "holding room" where they were given a participant information sheet (PIS) and completed a written informed consent form. They also completed pre- and post-participation questionnaires. Once enrolled, the TPs waited in the testing room (i.e., a different room to the practitioners), where they, too, were given a PIS and completed a written informed consent form and pre-participation questionnaires.

Aside from demographic information (e.g., gender, age, etc), the pre-participation questionnaires asked participants (both practitioners and TPs) to rate various characteristics on a 10 cm Visual Analog Scale (VAS). For example, both practitioners and TPs were asked to mark on the VAS their level of confidence in MRT in general, with the left of the VAS anchored with "No Confidence Whatsoever (0%)" and on the right with "Complete Confidence (100%)." They were also asked to rate their current level of test anxiety on a similar scale, with the left marked as "No Anxiety Whatsoever" and on the right with "Worst Anxiety Ever." In addition, before and after each round, participants were asked to rate other factors (e.g., how well do they know this person, how much do they like this person, how much connection do they feel with this person, etc.) on an ordinal scale of 0–10 (with 0 the lowest and 10 the highest).

In the testing room, seven complete testing stations were set up on 7 individual tables, evenly spaced around the room. Each testing station had a computer loaded with the research software, a keyboard, a mouse, an earpiece, and 2 chairs. The stations were configured in such a way that the TP could only see his/her monitor and no other monitors, and so that the practitioner could not see the TP's monitor. See Fig 1.

### 2.2. Test methods

Following the same basic methods as previous studies in this series, the primary index test, MRT, was used to detect deceit, and its results were compared to the reference standard of the actual verity of the spoken statement and the results of a secondary index test, intuitive guessing (i.e., without using MRT; IG). Each practitioner-TP pair performed 20 MRTs and 20 IGs, broken up into 2 blocks of each: 10 MRTs – 10 IGs – 10 MRTs – 10 IGs. Each practitioner tested all

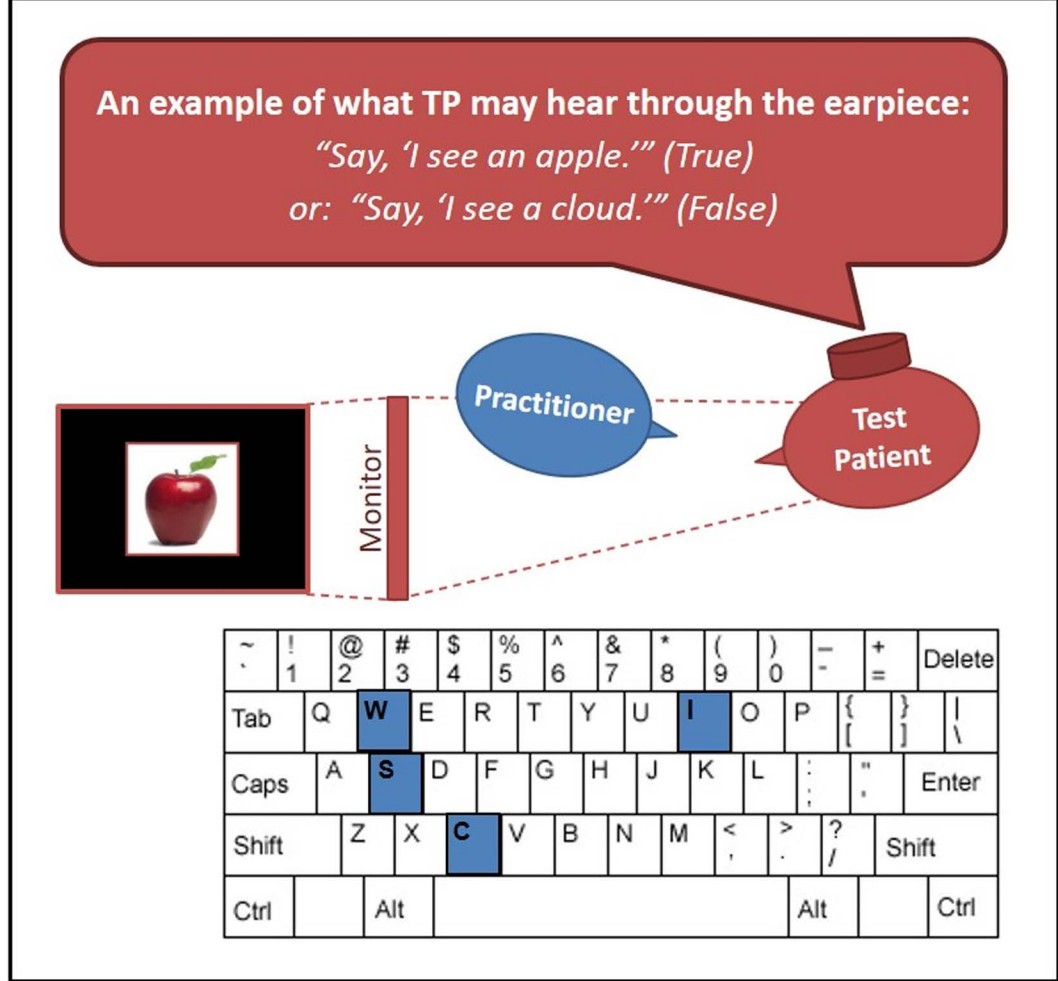

**Fig 1. Testing scenario layout.** The test patient (red) viewed a monitor which the practitioner could see, had an ear piece in his ear through which he received instructions. After the muscle test, the practitioner (blue) entered his results on a keyboard.

TPs and all TPs were tested by all practitioners. All practitioners were blind to the verity of the TP spoken statements in both conditions (MRT and IG).

The stimuli presented were selected from a database of 100 affect-neutral picture-statement pairs. DirectRT™ Research Software (Empirisoft Corporation, New York, NY) was programmed to randomly present a unique sequence of paired visual and auditory stimuli for each TP. The auditory stimuli were in the form of explicit instructions to the TP, such as, "Say, *'I see a ball.'*," and the sequence of true and false statements were randomised, with the prevalence of false statements kept constant at 0.50.

When testing began, the 7 TPs were seated at separate tables (see Fig 2). Practitioners entered the testing room and sat with a TP. Before starting, they each completed a number of pre-testing ratings about each other, and the practitioner could perform up to 5 practice MRTs. Then the testing began. As soon as the pair finished testing, they both completed a number of post-testing ratings about that round of testing, and then the practitioner moved on to another testing station and different TP. Once a practitioner completed testing all 7 TPs, s/he completed a post-testing questionnaire and

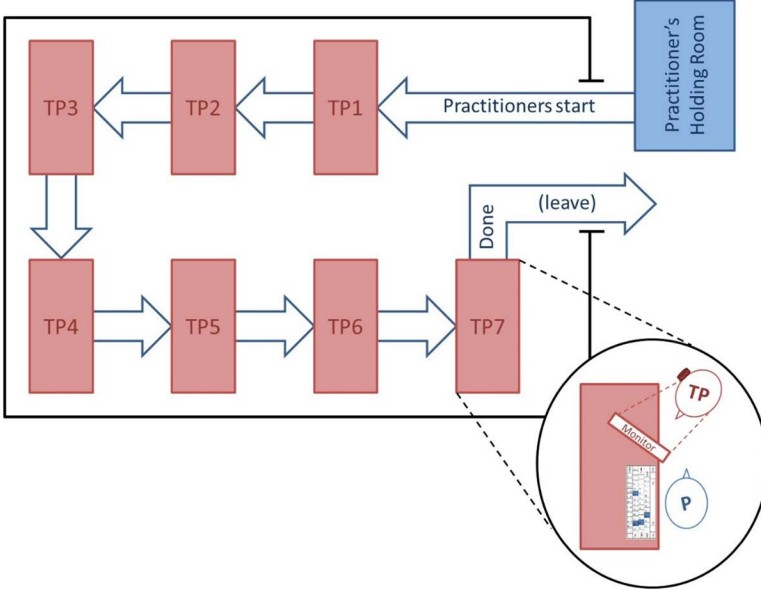

**Fig 2. Testing room flow.**

then was done with his/her participation. After each TP was tested by all 16 practitioners, they completed a short post-participation questionnaire and were also done with their participation. See Fig 3 for the Participant Flow Diagram.

### 2.3. Statistical methods

**2.3.1. Sample size.** A convenience sample of 7 test patients and 16 practitioners was recruited for pragmatic reasons, to make 112 patient/practitioner pairings. No additional sample size calculation was carried out, except to note that in previous studies in this series, statistical significance was reached using 20 and 48 patient/practitioner pairings.

**2.3.2. Methods of analysis.** For each pair, we calculated the following measures of accuracy for MRT and for IG: overall fraction correct, sensitivity, specificity, PPV and NPV [8] – and their 95% confidence intervals (95% CI). We calculated these across both blocks of tests (primary analysis) and separately for block 1 and block 2 (analysis of within-pairing repeatability). In addition, we report the mean measures grouped by both the practitioner and the TP. The same error-based measures will also be reported for IG.

Reproducibility, across TPs and across practitioners, were assessed by graphical methods were used for analysis. Following the recommendations of Bland and Altman we used both scatterplots and difference plots [18].

Repeatability between Block 1 and Block 2, within TP-practitioner pairing, was assessed similarly. Bland-Altman difference included reference lines at the mean difference, and 95% limits of agreement calculated as the mean difference ± 2SD. The smaller the range between these two limits (±2SDs) the better the agreement, and the less the bias. Bland-Altman plots of the difference between block 1 and block 2 scores (y-axis) against mean score (x-axis) are shown in S1 Fig (see S1A-F below).

Analysis of Variance (ANOVA) was used to determine how much of the variance in scores could be attributed to different models. One model looked at the influence of practitioners and TPs, and in another model, all participant characteristics were included (i.e., age, gender, years of experience, confidence, willingness, etc.).

All data were analyzed using Stata/IC 12.1 (StataCorp LP, College Station, Texas), specifically the commands "*twoway scatter*" and "*anova*."

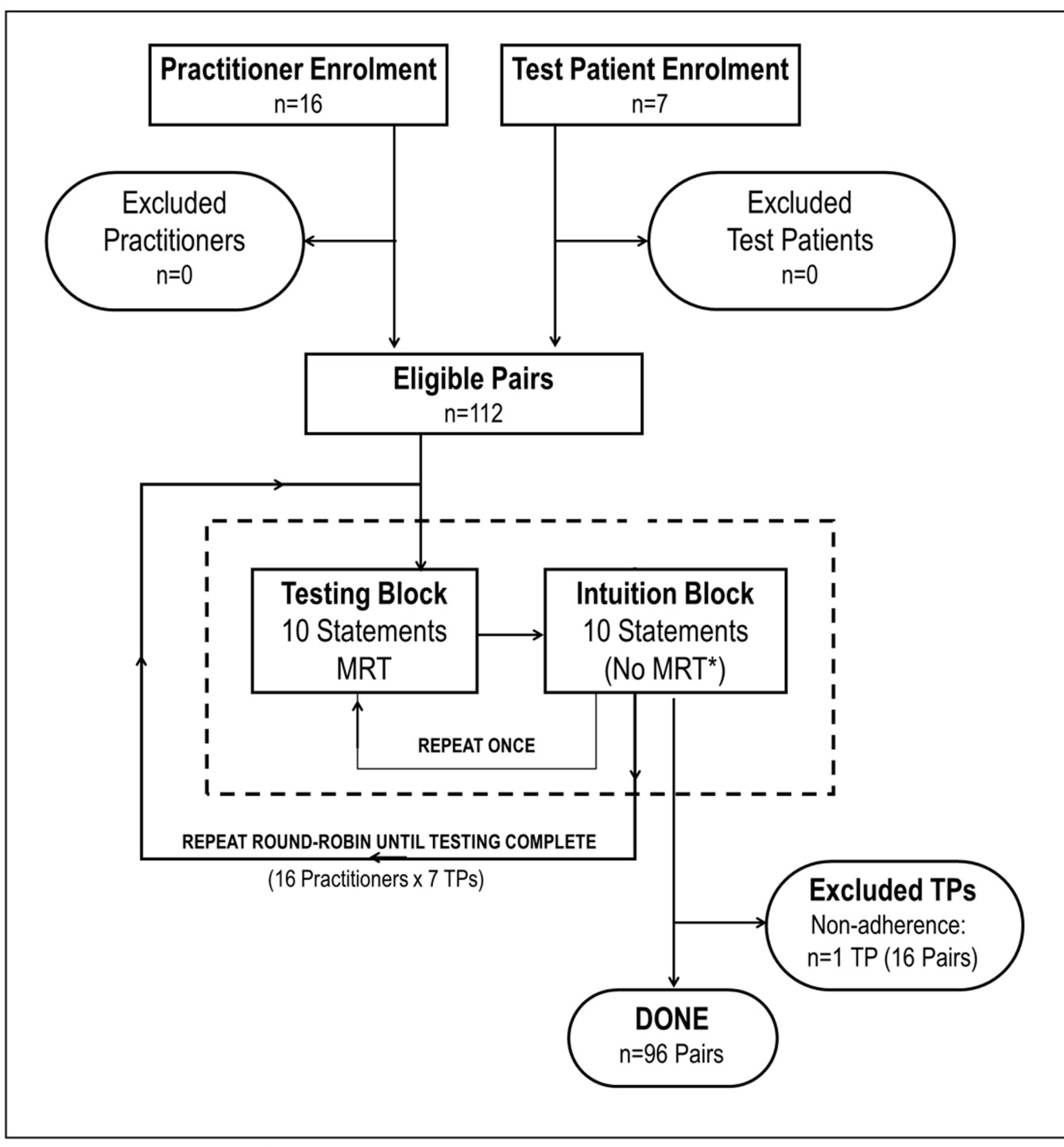

MRT, Muscle Response Testing; *Touching wrist & observing.

**Fig 3. Participant Flow Diagram.**

## 3. Results

### 3.1. Participants

Sixteen practitioners and 7 TPs were enrolled in late June 2012. Of the practitioners, 8 were male and 8 were female; 14 were chiropractors, 2 were acupuncturists; 9 were in full-time practice, 6 were in part-time practice and 1 was not currently practising; their mean (SD) age was 45.1 (12.4) years; and 15 reported being right-handed and 1, left-handed. Their mean (SD) number of years in practice was 13.8 (10.0) years, their mean (SD) number of years of using MRT in practice was 12.5 (8.8) years, and their mean usual hours/day using MRT was 5.5 (3.4) hours. Seven practitioners ranked their own MRT expertise as "4," seven ranked their own MRT expertise as "3," and two ranked their own MRT expertise as "1." No practitioners ranked their own MRT expertise as either "2" or "0." The mean (SD) score of self-ranked MRT expertise was 3.2 (0.7) out of a possible 4 (0="None" and 4="Expert"). As measured using the 10 cm Visual Analog Scales (VAS), their mean (SD) degree of confidence in own MRT ability (pre-participation) was 8.4 (1.5), their mean (SD) degree of confidence in MRT in general (pre-participation) was 8.0 (1.6) and their median degree of test anxiety (range) was 0.7 (0.0 to 5.0) out of a possible 10. For a summary of practitioner demographics, see Table 1. Also, for the Participant Flow Diagram, see Fig 3 (above).

One of the 7 TPs (originally labelled TP#5) failed to follow written and verbal instructions which resulted in no data being collected by her computer; therefore, she was excluded from all analyses. For convenience, the TP originally called TP#7 was renamed TP#5. Of the 6 remaining TPs, 3 were male and 3 were female; their mean (SD) age was 36.7 (15.0) years; and all 6 reported being right-handed. As measured using 10 cm VAS, their mean (SD; range) degree of experience with MRT ability was 5.3 (3.6; 0.2 to 9.9), their mean (SD) degree of confidence in MRT in general (pre- participation) was 8.0 (1.8) and their median degree of test anxiety (range) was 0.1 (0.0 to 3.5) out of a possible 10.

### 3.2. Test results

Participants took between 10 and 20 minutes to complete one round of testing. The duration of participation for practitioners was 1–1 ¼ hours, and the duration of participation for the TPs was approximately 3 hours, which including rounds of testing interspersed with short rest periods of 5–10 minutes. There were no adverse events reported from any testing.

**3.2.1. Accuracy.** Accuracy scores were calculated for each pair (n = 96 unique pairs), and mean accuracy scores were also calculated for each practitioner and each TP (n = 16 and n = 6 respectively; see S2 Table). For MRT, the mean overall accuracy (95% CI) was 0.616 (0.578–0.654), the mean sensitivity (95% CI) was 0.595 (0.549–0.640), the mean specificity (95% CI) was 0.638 (0.430–0.486), the mean Positive Predictive Value (PPV; 95% CI) was 0.632 (0.588–0.676), and the mean Negative Predictive Value (NPV; 95% CI) was 0.609 (0.5673–0.652). For IG, the mean overall accuracy (95% CI) was 0.507 (0.484–0.530), the mean sensitivity (95% CI) was 0.456 (0.424–0.487), the mean specificity (95% CI) was 0.557 (0.527–0.588), the mean PPV (95% CI) was 0.502 (0.475–0.530), and the mean NPV (95% CI) was 0.514 (0.491–0.537). In all these 5 measures of accuracy, MRT accuracy was found to be significantly more than both IG accuracy. See Table 2 below. Finally, there was no significant difference (p = 0.91) in mean MRT accuracies between those pairs containing a TP who reported guessing the paradigm (mean 0.609, 95% CI 0.191 to 1.000) and those pairs containing a TP who did not report guessing the paradigm (mean 0.623, 95% CI 0.397 to 0.849).

**3.2.2. Variability in MRT accuracy by practitioner and by test patient.** Fig 4A shows accuracy by practitioner, ordered from most accurate (left) to least accurate (right). The practitioner with highest accuracy, practitioner 8, had mean accuracy 79.2% (range 56.8% − 100.0%). The practitioner with lowest accuracy, practitioner 2, had mean accuracy 50.0% (range 39.0–61.0%). This difference by practitioner was statistically significant ($p < 0.001$ by likelihood ratio test in models adjusted for random effects by test patient).

Fig 4B shows accuracy by test patient, ordered by test patient number (left to right). The test patient with highest accuracy, test patient 6, had mean accuracy 73.8% (range 67.2% − 80.3%). The test patient with lowest accuracy, test patient 5,

**Table 1. Demographics of Practitioners.**

|  | Practitioners |
| --- | --- |
|  | (n = 16) |
| Gender (M:F) | 8:8 |
| Mean age (SD) | 45.1 (12.4) |
| Mean number of years in practice (SD) | 13.8 (10.0) |
| Practitioner-type (n) |  |
| Chiropractor | 14 |
| Acupuncturist | 2 |
| Practitioner Practice Status (n) |  |
| Full-time | 9 |
| Part-time | 6 |
| Not practising | 1 |
| Mean years of MRT experience (SD) | 12.5 (8.8) |
| Mean hours of MRT/day (SD) | 5.5 (3.4) |
| Mean self-ranked MRT Expertise* (SD) | 3.2 (0.7) |
| Self-ranked as "4" (n) | 7 |
| Self-ranked as "3" (n) | 7 |
| Self-ranked as "1" (n) | 2 |
| Median degree of test anxiety**† (Min, Max) | 0.7 (0.0, 5.0) |
| Mean degree of confidence in own MRT ability (pre-participation)† (SD) | 8.4 (1.5) |
| Mean degree of confidence in MRT in general (pre- participation)†(SD) | 8.0 (1.6) |
| Type(s) of MRT Technique(s) used (n)†† |  |
| Neuro Emotional Technique (NET) | 14 |
| Applied Kinesiology (AK) | 12 |
| Total Body Modification (TBM) | 2 |
| Contact Reflex Analysis (CRA) | 2 |
| NeuroModulationTechnique (NMT) | 2 |
| Other‡ | 6 |

MRT, Muscle Response Testing; SD, Standard Deviation; Min, Minimum; Max, Maximum; M, Male; F, Female; * Self-ranked MRT Expertise, ranged from 0 = None to 4 = Expert; ** Test Anxiety refers to the amount of anxiety the Practitioner was experiencing just prior to testing; †Measured using a 10 cm Visual Analog Scale, from 0 = "None" to 10="Most Ever"; †† Practitioners could respond with more than one technique; ‡ Other MRT techniques included 1 Practitioner each: Chiropractic Plus Kinesiology (CPK), Directional Non-Force Technique (DNFT), Jaffe-Mellor Technique (JMT), Lifeline, Nambudripad's Allergy Elimination Techniques (NAET), NeuroLink, Touch for Health.

**Table 2. Diagnostic Accuracy of MRT vs. IG.**

|  | MRT | | | Intuitive Guessing | | |  | p-value compared to Chance‡ | |
| --- | --- | --- | --- | --- | --- | --- | --- | --- | --- |
|  | n | Mean | 95% CI | n | Mean | 95% CI | p-value | MRT | Intuitive Guessing |
| **Accuracy*** | 96 | 0.616 | 0.578–0.654 | 96 | 0.507 | 0.484–0.530 | **<0.01†** | **<0.01†** | 0.53 |
| **Sensitivity** | 96 | 0.595 | 0.549–0.640 | 96 | 0.456 | 0.424–0.487 | **<0.01†** | **<0.01†** | **0.01†** |
| **Specificity** | 96 | 0.638 | 0.430–0.486 | 96 | 0.557 | 0.527–0.588 | **0.01†** | **<0.01†** | **<0.01†** |
| **PPV** | 96 | 0.632 | 0.588–0.676 | 96 | 0.502 | 0.475–0.530 | **<0.01†** | **<0.01†** | 0.87 |
| **NPV** | 96 | 0.609 | 0.5673–0.652 | 96 | 0.514 | 0.491–0.537 | **<0.01†** | **<0.01†** | 0.23 |

*Accuracy as Overall Fraction Correct; PPV, Positive Predictive Value; NPV, Negative Predictive Value; MRT, Muscle Response.

Testing; CI, Confidence Interval; ‡ 50−50 Chance; †Significance reached.

Means, 95% and Significance. Accuracy (Overall Fraction Correct), Sensitivity, Specificity, Positive Predictive Value, and Negative Predictive Value.

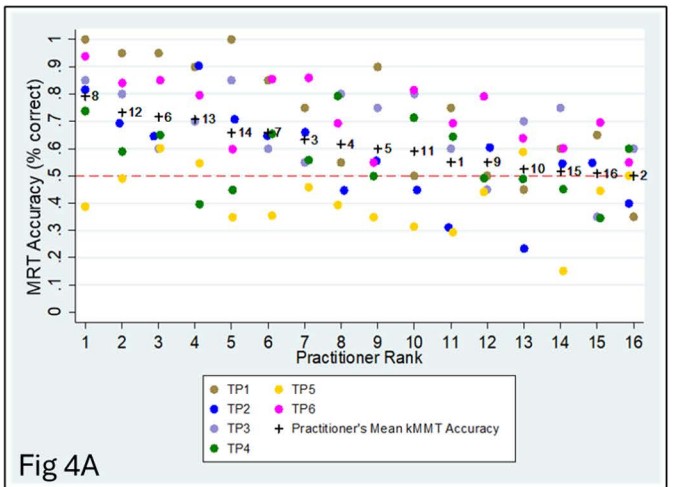
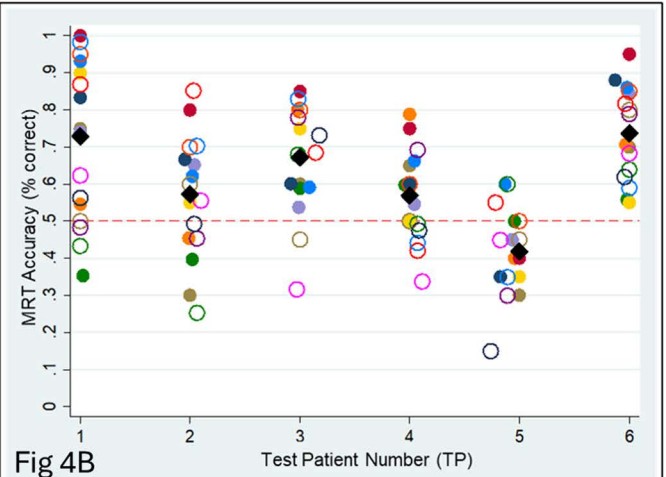

**Fig 4. Reproducibility.** Scatterplot of MRT Accuracy vs. Participant. (A) by Practitioner, (B) by TP.

**Table 3. ANOVA results for Practitioner and Test Patient.**

| Source | Partial SS | df | F | Prob>F | % |
|---|---|---|---|---|---|
| **Model** | 1.8963 | 20 | 4.97 | <0.01 | **57.0%** |
| **Practitioner** | 0.7196 | 15 | 2.51 | <0.01 | **21.6%** |
| **Test Patient** | 1.1767 | 5 | 12.33 | <0.01 | **35.4%** |
| **Residual** | 1.4312 | 75 | | | **43.0%** |
| **Total** | 3.3275 | 95 | | | 100.0% |

had mean accuracy 41.9% (range 35.5% − 48.2%). This difference by patient was statistically significant ($p<0.001$ by likelihood ratio test in models adjusted for random effects by practitioner).

In two-way mixed effects ANOVA (Table 3), practitioner explained 21.6% of variation in accuracy ($p<0.001$) and test patient explained 35.4% of variation in accuracy ($p<0.001$).

**3.2.3. Repeatability.** Repeatability was considered to be the amount of variance in the MRT accuracy between block 1 and block 2 for each pair. It can be reported by practitioner or by TP. In tests with good repeatability, scatterplots of block 1 scores vs. block 2 scores will hover along a diagonal line (slope = 1) indicating identical scores.

For scatterplots of mean MRT accuracies by practitioner (A) and by TP (B), see Fig 5 (below). Visual inspection of these scatterplots suggests a reasonable amount of agreement between mean MRT accuracies in Blocks 1 and 2, especially with respect to TPs (B), which suggests adequate repeatability.

## 4. Discussion

### 4.1. Statement of principal findings

**4.1.1. Reproducibility and repeatability.** The average MRT accuracy was 61.6% (95% CI 57.8% − 65.4%), but some pairs surpassed this with accuracies of over 80%. Certain TPs seemed "easier" to test, evidenced by higher mean accuracies for pairs including them. For instance, TP#1 yielded a mean accuracy of 72.8% (range 61.3% – 84.3%), and two practitioners ([8] and [14]) achieved a perfect score of 100% accuracy (see S2 Table). In contrast, TP#5 posed more challenges, with a mean accuracy of just 41.9% (range 35.5% − 48.2%). The reason for this discrepancy requires further

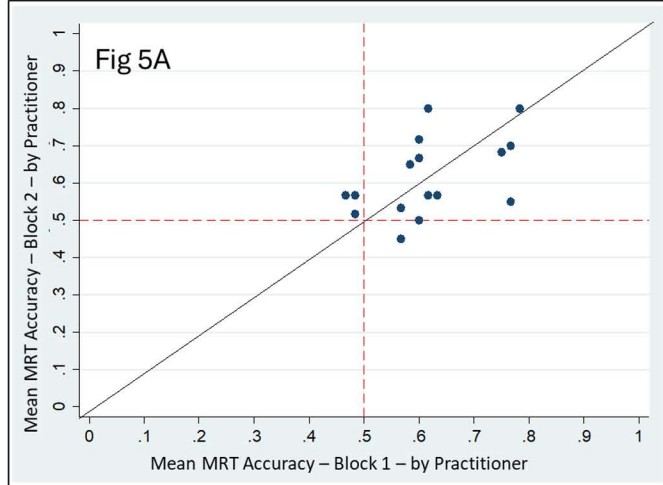
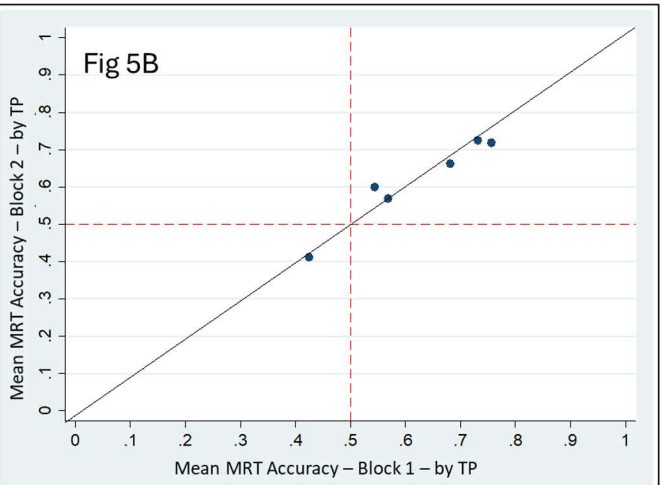

**Fig 5. Repeatability Scatterplots.** Mean MRT Accuracy – Block 1 vs Block 2. (A) by Practitioner, (B) by TP.

investigation. The definitive implications and relevance of these results, however, couldn't be ascertained within this study's scope.

The high repeatability within-practitioner/TP pair (i.e., between Blocks 1 and 2) suggests the practitioner effects (and patient effects) are sufficiently repeatable. In other words, the intra-tester variability is low, indicating that the differences in accuracy between practitioners and TPs are more likely due to genuine differences in skill or testing conditions rather than random variations or errors.

These results indicate that the high level of accuracy observed in certain practitioners, once if better understood, may be able to be replicated through training across the broader practitioner community, potentially raising the standard of proficiency in the field.

**4.1.2. MRT accuracy vs. IG accuracy.** Supporting results from previously reported studies in this series, MRT once again was found to distinguish lies from truth significantly more accurately than either chance or IG. Also, the finding that the mean specificity, 0.638 (95% CI 0.430 to 0.486), was more than the mean sensitivity, 0.595 (95% CI 0.549 to 0.640), suggests that truths were generally easier to detect than lies; however, the difference between these means did not reach significance ($p = 0.1759$), so caution is advised when interpreting these results.

Only 57% of the between-pairs variance in mean MRT accuracy could be explained by the pair itself, leaving 43% attributable to unknown factors. The within-pair variance of mean MRT accuracy suggests adequate repeatability. In addition, visual inspection of reproducibility and repeatability scatterplots and ANOVA results seems to suggest that stability of MRT accuracy may be more TP-specific than practitioner-specific. The results for each practitioner, individually, suggest that both reproducibility and repeatability may be sufficiently stable for some practitioners, and insufficient for others. The results for each TP, individually, suggest the same. Overall, analyses of both reproducibility and repeatability of MRT accuracy showed variances that could not be explained by any single factor.

## 4.2. Comparisons to other studies

There are a number of studies published in the muscle testing literature which have attempted to quantify the reliability of muscle testing used in various capacities. Using another type of MMT, namely Applied Kinesiology style of MMT (AK-MMT), Conable reports finding fair intra-tester agreement ($\kappa = 0.54$) [19], Perot et al. report finding good reliability [20], and Pollard et al. report finding good agreement between a novice practitioner and an experienced practitioner when

using AK-MMT [21]. In 2 studies also using AK-MMT mixed inter-examiner agreement was found [22,23]. On the other hand, systematic reviews of AK-MMT found that its reliability could not be adequately determined [24–26], and in two studies using MRT both intra-tester and inter-tester reliability were found to be insufficient [27,28]. The mixed results of these reports, along with the findings of this present study, speak to the difficulty of exploring the variability of AK-MMT accuracy.

### 4.3. Strengths and limitations

A strength of this study is that its results support the findings of earlier studies, demonstrating that MRT accuracy can be adequately estimated using rigorous scientific methods and that these methods produce durable results. Other strengths are the breadth of the types of the MRT-practitioners enrolled, and the degrees of experience of all participants. In addition, similar to other studies in this series, testing during this study was as true to clinical practice as possible in a research setting. A limitation of this study is its generalisability to other applications of MRT, to MRT using muscles other than the deltoid, and to other forms of MMT (such as to AK-MMT). Also, a weakness in the study design may have been the duration of participation of the TPs: 3 + hours in total. This amount of time may have adversely impacted their performance or their compliance to strict procedures.

### 4.4. Implications for clinical practice

Naturally, a clinician would like to know how stable or reliable MRT accuracy is. S/He would be interested in knowing if it can be relied upon from one patient to the next (reproducibility) and with the same patient from one visit to the next (repeatability). Therefore, the primary concern is whether the largest variability is small enough to be clinically meaningful [29], and the results of this study suggest that this must be taken on a pair-by-pair basis. On the other hand, how reliable MRT accuracy must be in order for it to be useful is a clinical decision, not a statistical one [30].

### 4.5. Unanswered questions and future research

The results of this study suggest that the accuracy of MRT used in this way may be sufficiently reproducible and repeatable. However, since MRT is a test used by practitioners to guide treatment usually within the context of a specific protocol or technique system, to assess the true usefulness of MRT, randomised, controlled trials must be carried out to assess the effectiveness of the various technique systems that employ MRT. In other words, aside from MRT being accurate or precise in and of itself, it must be ascertained if its use leads to improved patient outcomes, such as a better quality of life. Future research must focus on responding to this concern.

## 5. Conclusion

The consistency of MRT accuracy achieved by individual practitioners across various TPs, as well as the uniformity of MRT accuracy observed with a single TP across multiple practitioners, suggests a replicable level of proficiency in the application of this technique. Furthermore, the existence of unexplained factors beyond the scope of the current model indicates the necessity for additional research to fully understand and account for the observed variance.

## Supporting information

**S1 Fig. Bland-Altman Plots.** Bland-Altman Plots of the difference between block 1 and block 2 scores (y-axis) against mean score (x-axis).
(TIF)

**S1 Table. Mean Accuracy Data for each Practitioner and each Test Patient individually. Accuracy (Overall Percent Correct), Sensitivity, Specificity, PPV and NPV; for MRT and Intuition.**
(PDF)

**S2 Table. Mean Accuracy Data for each Practitioner and each TP individually. Accuracy (Overall Percent Correct), Sensitivity, Specificity, PPV and NPV; for MRT and Intuition.**
(XLSX)

**S3 Table. ANOVA results for Practitioner, Test Patient and Block.**
(XLSX)

**S1 Dataset. Raw, deidentified data used in the analysis of muscle response testing accuracy. Includes practitioner demographics, professional experience, and trial-level outcomes across multiple participants. Variable names and coding definitions are provided in the dataset columns.**
(XLSX)

## Acknowledgments

The authors would like to thank those who volunteered their time to participate in this study.

## Author contributions

**Conceptualization:** Anne Marie Jensen, Richard J. Stevens.

**Data curation:** Anne Marie Jensen, Richard J. Stevens.

**Formal analysis:** Anne Marie Jensen, Richard J. Stevens.

**Investigation:** Anne Marie Jensen.

**Methodology:** Anne Marie Jensen, Amanda J. Burls, Richard J. Stevens.

**Project administration:** Anne Marie Jensen.

**Software:** Anne Marie Jensen.

**Supervision:** Amanda J. Burls, Richard J. Stevens.

**Validation:** Anne Marie Jensen.

**Writing – original draft:** Anne Marie Jensen.

**Writing – review & editing:** Anne Marie Jensen, Richard J. Stevens.

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
