## [Decision Letter · Decision Letter 0]

Exploring the variation in muscle response testing accuracy through repeatability and reproducibility

PLOS ONE

Dear Dr. Jensen,

Thank you for submitting your manuscript to PLOS ONE. After careful consideration, we feel that it has merit but does not fully meet PLOS ONE’s publication criteria as it currently stands. Therefore, we invite you to submit a revised version of the manuscript that addresses the points raised during the review process.

We look forward to receiving your revised manuscript.

Kind regards,

Mário Espada, PhD

Academic Editor

PLOS ONE

Journal Requirements:

3. Please amend your manuscript to include your abstract after the title page.

Additional Editor Comments:

Dear Authors,

Please consider the minor revisions suggested by reviewer 1.

Thank you.

Best regards.

Reviewers' comments:

Reviewer's Responses to Questions

**Comments to the Author**

1. Is the manuscript technically sound, and do the data support the conclusions?

Reviewer #1: Partly

Reviewer #2: Yes

2. Has the statistical analysis been performed appropriately and rigorously?

Reviewer #1: Yes

Reviewer #2: Yes

3. Have the authors made all data underlying the findings in their manuscript fully available?

Reviewer #1: Yes

Reviewer #2: Yes

4. Is the manuscript presented in an intelligible fashion and written in standard English?

Reviewer #1: Yes

Reviewer #2: Yes

Reviewer #1: The authors of the current manuscript utilized a novel experimental design and statistical analyses to investigate both repeatability and reproducibility of muscle response testing (MRT) compared to that of intuitive guessing. Based upon the results of the study, MRT appeared to be more accurate—evidenced by greater repeatability and reproducibility—compared to intuition alone. These were interesting findings, and the authors did seem to utilize sound experimental design and rigorous statistical analyses to obtain these findings.

Specific Comments:

- The authors did highlight that certain test participants produced a greater degree of variability. This may be beyond the scope, but are there any other statistical analyses that could be performed to tease out potential correlational relationships between other variables (e.g., gender, age, answers provided on pre- and post-tests)? There is no basis to suggest this, however as an example perhaps greater TP variability correlated with pre-test anxiety scores etc. Perhaps I missed these analyses within the results section, however I did read the manuscript carefully and thoroughly.

- The methods section could be improved with a more detailed explanation of the MRT techniques and tests, specifically. Perhaps this is referenced elsewhere, although I was unable to find detailed explanation of the MRT protocols within the methods section.

- In line with the previous point, I think a standardized testing methodology for the MRT would be of benefit. For example, the deltoid muscle was used for the MRT procedure; the practitioner assessed the muscle for precisely “x” seconds etc. Perhaps this is not how MRT protocols work, but it would be of great benefit for the purpose of reproducibility by outside research groups.

Reviewer #2: The paper by Jensen et al. is interesting and very well written.

MRT and MMT are not completely defined, assuming the reader knows them. And yet when discussing Specificity, Sensibility and other characteristics, the gold standard is not clear for the reader. This is the only observation this reviewer would make, that it would be useful for a better comprehension to clarify what the misses and the successes consist of, when evaluating binary results.

Line 28: the word "and" is to be deleted.

**Do you want your identity to be public for this peer review?** For information about this choice, including consent withdrawal, please see our Privacy Policy

Reviewer #1: No

Reviewer #2: No

---

## [Author Response · Author response to Decision Letter 1]

2 May 2025

See also our latest Response to Reviewers document - previously uploaded.

This is our Response to Editor's Requests:

Comment 1. We note that several of your files are duplicated on your submission. Please remove any unnecessary or old files from your revision, and make sure that only those relevant to the current version of the manuscript are included.

Response: My apologies – I will remove old, unnecessary files and resubmit.

Comment 2. Please include a complete copy of PLOS’ questionnaire on inclusivity in global research in your revised manuscript.

Response: Okay – it’s now included.

Comment 3. Can you please upload an additional copy of your revised manuscript that does not contain any tracked changes or highlighting as your main article file.

Response: Okay – I tried to do this before. So I tried again.

Comment 4. Please amend your manuscript to include your abstract after the title page.

Response: Done.

Comment 5. Please include your tables as part of your main manuscript and remove the individual files. Please note that supplementary tables (should remain/ be uploaded) as separate "Supporting Information" files

Response: Our 3 tables were included in the main manuscript. Then we also included 3 Supplementary Tables (and Supplementary Figures) which we uploaded as Supporting Information in a ZIP folder. Would you like us NOT to use a ZIP folder?

---

## [Decision Letter · Decision Letter 1]

Exploring the variation in muscle response testing accuracy through repeatability and reproducibility

PONE-D-24-53858R1

Dear Dra. Anne Marie Jensen,

We’re pleased to inform you that your manuscript has been judged scientifically suitable for publication and will be formally accepted for publication once it meets all outstanding technical requirements.

Kind regards,

Mário Espada, PhD

Academic Editor

PLOS ONE

Reviewers' comments:

Reviewer's Responses to Questions

**Comments to the Author**

Reviewer #1: All comments have been addressed

2. Is the manuscript technically sound, and do the data support the conclusions?

Reviewer #1: Yes

3. Has the statistical analysis been performed appropriately and rigorously?

Reviewer #1: Yes

4. Have the authors made all data underlying the findings in their manuscript fully available?

Reviewer #1: Yes

5. Is the manuscript presented in an intelligible fashion and written in standard English?

Reviewer #1: Yes

Reviewer #1: (No Response)

---

## [Editor Report · Acceptance letter]

PONE-D-24-53858R1

PLOS ONE

Dear Dr. Jensen,

I'm pleased to inform you that your manuscript has been deemed suitable for publication in PLOS ONE. Congratulations! Your manuscript is now being handed over to our production team.

Kind regards,

on behalf of

Dr. Mário Espada

Academic Editor

PLOS ONE